# Optimization of the Core Compound for Ytterbium Ultra-Short Cavity Fiber Lasers



Andrey Rybaltovsky [1], Mikhail Yashkov [2], Alexey Abramov [2], Andrey Umnikov [2], Mikhail Likhachev [1] and Denis Lipatov [2,*]

[1] Prokhorov General Physics Institute of the Russian Academy of Sciences,
Dianov Fiber Optics Research Center, 119333 Moscow, Russia; rybaltovsky@yandex.ru (A.R.);
likhachev@fo.gpi.ru (M.L.)

[2] G. G. Devyatykh Institute of Chemistry of High-Purity Substances, Russian Academy of Sciences,
St. Tropinina 49, 603951 Nizhny Novgorod, Russia; yashkovmv@yandex.ru (M.Y.);
abramovan84@mail.ru (A.A.); umkand@yandex.ru (A.U.)

[*] Correspondence: lidenis@yandex.ru; Tel.: +7-951-907-5010

**Abstract:** Highly ytterbium-, aluminum- and phosphorus-co-doped silica fibers with low optical losses were fabricated by the MCVD method, utilizing an all-gas-phase deposition technique. Optical and laser properties of the active fibers with a phosphosilicate and aluminophosphosilicate glass cores doped with 1.85 mol% and 1.27 mol% $Yb_2O_3$ were thoroughly investigated. With the help of hydrogen loading, it was possible to induce highly reflective Bragg grating in both fiber samples using the standard phase-mask technique and 193 nm-UV laser irradiation. The ultra-short (less than 2 cm long) Fabry–Perot laser cavities were fabricated by inscribing two fiber Bragg gratings (highly and partially reflective FBGs) directly in the core of the fiber samples. The highest pump-to-signal conversion efficiency of 47% was demonstrated in such laser configuration using phosphosilicate fiber. The reasons for the low efficiency of aluminophosphosilicate fiber are discussed.

**Keywords:** ytterbium-doped fiber; phosphosilicate glass; aluminophosphosilicate glass; fiber Bragg grating; ultra-short cavity fiber laser; single-frequency fiber laser

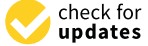



## 1. Introduction

In recent years, single-frequency fiber lasers (SFLs) have been increasingly used in high-speed telecommunications, spectroscopy, sensors and metrology. Due to their compactness, reliability and simple design, these devices can be easily integrated into existing fiber optic systems.

The cavity of the SFL is a short (only a few centimeters long) segment of an active fiber located between two wavelength-matched Bragg gratings (FBGs) [1] FBGs in a fiber cavity serve the same function as reflective mirrors in a classic Fabry–Perot cavity: They provide feedback. Due to the small (several millimeters) distance between the FBGs in a short fiber Fabry–Perot cavity, stable selection of a single longitudinal mode is ensured, which leads to a single-frequency laser operation regime.

One of the most relevant and promising applications of SFLs generating radiation at a wavelength of ~1.06 μm is the circuits of nonlinear optical frequency conversion. Based on such schemes, high-power laser sources of highly coherent radiation in the visible and ultraviolet (UV) wavelength ranges have been created, successfully competing with gas and solid-state lasers [2,3].

To achieve single-frequency operation, it is necessary to create a very short fiber laser cavity of a few centimeters in length to keep only one longitudinal mode within the FBG reflection spectrum. In turn, to achieve lasing for such a short resonator, it is necessary to use fibers with the maximum possible concentration of ytterbium ions in the core. The best efficiency of 72.7% to date was achieved using phosphate glass fibers [4]. However, this

type of fibers has well-known drawbacks due to its low thermal and mechanical stability. One possible solution to this problem could be the utilization of composite fibers with phosphate core and silica cladding [5], but to date, the maximum concentration of Yb demonstrated in such fibers was much smaller as compared to phosphate fibers due to a strong diffusion between core and cladding. The pump-to-signal conversion efficiency of the SFL fabricated on the Yb-doped composite fiber was ~10% at an operation wavelength of 1030 nm [6].

For silica-based fibers, the maximum pump-to-signal conversion efficiency of ~20% in SFLs operated near 1 μm was reported in [7]. Photosensitive silica-based fiber doped with 0.84 mol% $Yb_2O_3$ in the core was utilized in this case to write π-shifted high-reflection FBG.

It should be noted that a much higher Yb concentration could be incorporated into silica-based fibers without loss of active properties [8,9]. In [8], three types of Yb-doped fibers were studied—phosphosilicate, aluminosilicate and aluminophosphosilicate. It was shown that aluminosilicate and aluminophosphosilicate fibers suffer from concentration quenching—fibers nearly completely lose their active properties, when $Yb_2O_3$ concentration reaches 2 mol% [8], while phosphosilicate fibers were free from this effect. No pump-to-signal efficiency degradation was observed in phosphosilicate fibers up to 2.5 mol% of $Yb_2O_3$ (maximal concentration, which could be incorporated using MCVD technology). An important advantage of the phosphosilicate Yb-doped fibers is also the highest resistance to photodarkening under prolonged exposure to pump radiation (λ~0.98 μm) [9]. The feature of this type of fibers is a relatively high difference between the refractive indices of the core and cladding ($\Delta n_{core-clad}$) reaching ~0.02 or more. It is coursed by a high molar refractivity of $Yb_2O_3$ ($5.4 \times 10^{-3}$ $mol^{-1}$) [10] and a high concentration of $P_2O_5$ (~10 mol%) required to achieve good solubility of $Yb_2O_3$ (molar refractivity of $P_2O_5$ (~$0.88 \times 10^{-3}$ $mol^{-1}$) [11]. High $\Delta n_{core-clad}$ is helpful to reduce lasing threshold due to increased pump power density in the core as compared to standard fibers. All these features make phosphosilicate fibers to be a promising candidate for building ultra-short laser cavities.

The main disadvantages of the phosphosilicate fibers are a small absorption and emission cross-section, which are two times smaller than in the aluminosilicate fibers. As a result, aluminophosphosilicate fiber with a concentration of only 1.27 mol% of $Yb_2O_3$ demonstrates a high pump-to-signal conversion efficiency of more than 70% that could be achieved for a record short (less than 4 cm) silica-based fiber amplifier. This feature makes aluminophosphosilicate fiber to be another promising candidate for the role of active medium of SFLs based on silica-based fibers. In case the content of oxides $Al_2O_3$ and $P_2O_5$ in the glass is equal, they are completely bound into $AlPO_4$ structural groups (joint), as a result of which aluminophosphosilicate glass has the same refractive index as undoped quartz glass [12]. The increase in the numerical aperture in the Yb-APS fiber core is proportional to the concentration of $Yb_2O_3$ in the glass.

The laser efficiency and stability of single-frequency fiber lasers are defined by the length of its laser cavity—to obtain only one longitudinal mode it should be short enough (typically less than 2 cm), but to achieve higher efficiency of the pump absorption in such a resonator, it must be as high as possible. Thus, the main parameters of single-frequency lasers depend critically on the parameters of the used active fiber. In this work, the detailed comparative study of optical characteristics of short Fabry–Perot laser cavities operating in the vicinity of ~1.06 μm and based on silica-based fibers doped with a maximum possible ytterbium concentration was performed for the first time. We have considered the two most promising core glass compositions—$Yb_2O_3/P_2O_5/SiO_2$ and $Yb_2O_3/Al_2O_3/P_2O_5/SiO_2$.

## 2. Materials and Methods

The main purpose of the research was to create short-length laser cavities, taking into account that their output characteristics principally depend on the properties of the optical fibers.

### 2.1. Optical Fiber Preforms Preparation Details

The fiber preforms were fabricated by the fully gas-phase-modified chemical vapor deposition (MCVD) technology, using liquid volatile compounds $SiCl_4$, $POCl_3$, $CCl_4$, $C_2F_3Cl_3$ and low-volatile solid powders $AlCl_3$ and $Yb(thd)_3$ (thd = 2,2,6,6-tetramethyl-3,5-heptanedicionate). Containers with powder precursors were heated to 130–160 °C, and the vapors were delivered into the deposition tube through heated lines. It should be noted that, under these conditions, the vapor pressure of $AlCl_3$ was comparable to that of liquid chlorides, in contrast to $Yb(thd)_3$, whose pressure was two orders of magnitude lower than that of other precursors even at the maximum possible thermostating temperature of ~165 °C (limited by the powder melting point). In order to introduce a higher concentration of $Yb_2O_3$ into the core, we used several containers with $Yb(thd)_3$, i.e., multiplied the source of ytterbium.

The core glass was fabricated using an original technique for the separate deposition of glass components [8,13]. A scheme for the manufacturing process of an aluminophosphosilicate core doped with $Yb_2O_3$ is depicted in Figure 1. In the first stage, at a low temperature of 1350–1550 °C, a partially fused layer of the phosphosilicate or aluminophosphosilicate matrix was deposited. In the second stage, $Yb(thd)_3$ vapors were mixed with oxygen and delivered into the deposition tube, and $Yb_2O_3$ particles formed in the hot zone (1000–1100 °C) were deposited on the surface of the matrix layer (gas phase impregnation). In the third stage, the $Yb_2O_3$ layer/matrix was sintered into a transparent glass (1850–1950 °C) in an atmosphere of $CCl_4$ and oxygen to remove water impurities formed at the impregnation stage. The core was made by the deposition of several thin layers of the matrix, each of which was impregnated with $Yb_2O_3$ for a long enough time (16–18 main burner passes).

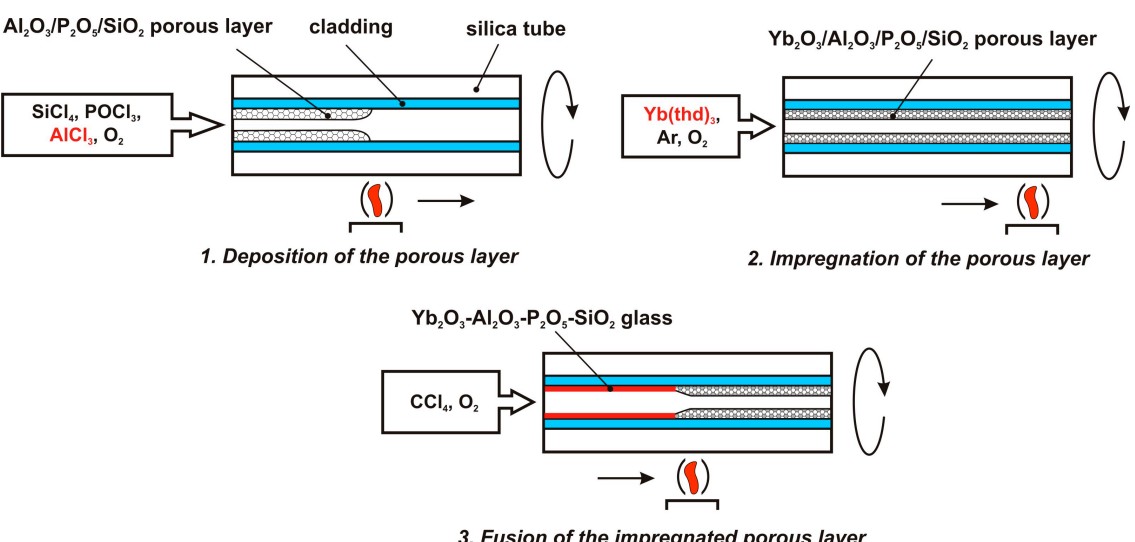

**Figure 1.** Scheme of the original core fabrication technique with separate deposition of glass components.

This technique provides potentially any concentration of the active additive in the glass by increasing the impregnation time of a single layer or the number of corresponding impregnation passes. The conversion of precursors proceeds at a significantly lower temperature relative to the standard MCVD technique. This ensures the homogeneity of the glass composition along the length of the preform (premature precipitation of $Yb_2O_3$ before the burner is excluded), a high concentration of phosphorus in the glass (since $P_2O_5$ is volatile under the conditions of the MCVD method) and a low concentration of hydroxyl groups in the synthesized glass (sintering of the impregnated glass layer proceeds in the absence of a large amount of $H_2O$, a source which is $Yb(thd)_3$).

As is known, the concentration of $Yb_2O_3$ in silicate glasses is determined by the amount of co-dopants. In the case of Yb-doped phosphosilicate core glass (Y291 preform

sample), we managed to introduce 12.5 mol% $P_2O_5$ and 1.85 mol% $Yb_2O_3$ (Table 1). The concentration of dopants was limited by the formation of bubbles during the sintering process of the layers, due to the joint evaporation of phosphorus and ytterbium oxides. The Yb-doped aluminophosphosilicate core glass (LD604 preform sample), as shown in Table 1, contained 9 mol% $Al_2O_3$, 10.2 mol% $P_2O_5$ and 1.27 mol% $Yb_2O_3$. A further increase in the concentration of dopants led to the cracking of the synthesized glass due to the large difference between the coefficient of thermal expansion (CTE) of the core and cladding materials.

**Table 1.** Fiber preforms core glass chemical composition.

| Preform # | Core Class | $Yb_2O_3$ Content, mol% | $Al_2O_3$ Content, mol% | $P_2O_5$ Content, mol% |
|-----------|------------|-------------------------|-------------------------|------------------------|
| LD604 | $Yb_2O_3/Al_2O_3/P_2O_5/SiO_2$ | 1.27 | 9 | 10.2 |
| Y291 | $Yb_2O_3/P_2O_5/SiO_2$ | 1.85 | – | 12.5 |

*2.2. Optical Fiber Samples Properties*

Single-mode fibers with an outer diameter of 125 μm were drawn from the fabricated preforms. The most significant measured parameters of the drawn fiber samples (LD604 and Y291) are given in Table 2. It is well known that Yb-co-doped phosphosilicate fibers optical losses in the near-infrared (NIR) spectral range are mainly originated from absorption band of $Yb^{3+}$ ions at 975 nm [14]. Since the absorption at a wavelength of 975 nm is typically pretty high, this wavelength is most often used to pump ytterbium-doped lasers operating in the vicinity of ~1.06 μm. The cutoff wavelength in both fibers (at 850 and 910 nm, respectively) is shorter than the pump and the lasing wavelengths, which is important for achieving optimal laser efficiency [15]. In Yb-co-doped fibers, the level of background losses can be fairly estimated at the wavelength of ~1.55 μm, where the contribution of ytterbium dopant absorption bands is obviously negligible. As it can be seen from Table 2, the level of background losses in both studied fibers does not exceed 0.01 dB/m, which is a sign of homogeneously produced preforms and high purity of our fiber core glasses (actually, it is an indicator that harmful impurities of transition metals and heavy rare-earth clusters are absent).

**Table 2.** Fiber samples optical characteristics.

| Fiber # | $\Delta n_{core-clad}$, RIU | $\lambda_{cut-off}$, nm | Loss at 975 nm, dB/m | Loss at 1550 nm, dB/m |
|---------|------------------------------|--------------------------|----------------------|------------------------|
| LD604 | 0.01 | 850 | 2600 | 0.01 |
| Y291 | 0.018 | 910 | 1700 | 0.01 |

*2.3. Fiber Laser Ultra-Short Cavities Fabrication Technique*

Two Fabry–Perot fiber laser (also called a "Distributed Bragg Reflector" or "DBR" laser) cavities studied in this work were formed entirely on short segments (~2 cm length long) of the active fibers LD604 and Y291, respectively. The FBGs inscription technique was carried out using a standard technique of UV laser irradiation through a phase mask. Since phosphosilicate fibers have relatively low initial photosensitivity to UV radiation, inscribing of FBGs inside them is possible only in the presence of molecular hydrogen (with a concentration of at least $10^{19}$ $cm^{-3}$) in the core region [16]. Like phosphosilicate, rare-earth doped aluminosilicate fibers have also sufficient UV photosensitivity only after hydrogenation treatment [17]. Therefore, the segments of active fibers used for FBGs inscribing were preliminarily kept in an atmosphere of molecular hydrogen at room temperature ($\approx$295 K) and pressure of 10 MPa for approximately 3 weeks to be sure the core is completely saturated with hydrogen [18]. Using a phase mask with a period of 732 nm and a 193 nm wavelength pulsed radiation (pulse duration of ~10 ns and repetition rate of 10 Hz) of the Optosystems CLS-5000 ArF excimer laser, two wavelength-matched uniform

FBGs (highly reflecting or "HR" and partially reflecting or "PR") of the same length (8 mm) were induced in the core of the active fiber at a distance of 2 mm from each other. Thus, the total length of each fabricated laser cavity ("Laser #1" and "Laser #2"), including the HR and PR FBGs, was 18 mm.

The transmission spectra of FBGs corresponding to Laser #1 and Laser #2 were measured immediately after their inscription using the Yokogawa AQ6370C optical spectrum analyzer, and the spectra are depicted in Figure 2. As can be seen from this figure, the maximum intensity of the Bragg peaks of each of the FBGs is placed near the wavelength of 1066 nm. Both HR and PR FBG pairs (curves 1 and 2, 3 and 4) have good spectral overlapping, and relevant peaks centered almost identically.

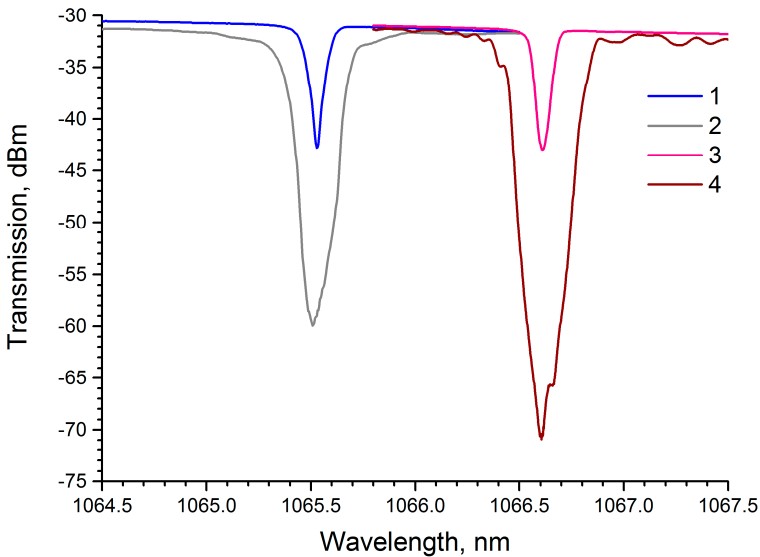

**Figure 2.** Transmission spectra of FBGs inscribed in short segments of the fiber samples LD604 (curves 1, 2) and Y291 (curves 3, 4).

The reflection coefficients (reflectance) of the HR and PR FBGs are shown in Table 3 for both Laser #1 and Laser #2 cavities. According to the analytical model of a short Fabry–Perot cavity from [19] and using equations from [20], the effective cavity lengths of Laser #1 and Laser #2 were calculated with the same result of ~5 mm. Based on this value, the longitudinal mode-spacing was obtained as about 0.08 nm or 9.5 GHz. It is also important to compare the total UV irradiation dose spent on inducing similar FBGs in the LD604 and Y291 fiber segments: For HR FBGs, the irradiation dose values differ by a factor of 6.2, and for PR FBGs, by a factor of 4.5. This means that under the same UV irradiation conditions, fiber LD604 has much weaker photosensitivity than fiber Y291.

**Table 3.** Laser cavity samples and associated FBGs characteristics.

| Laser # | Fiber # | Reflectance, % | | Irradiation Dose, J/cm$^2$ | |
|---|---|---|---|---|---|
| | | **HR FBG** | **PR FBG** | **HR FBG** | **PR FBG** |
| 1 | LD604 | 99.9 | 93.2 | 3000 | 480 |
| 2 | Y291 | 99.99 | 93.1 | 540 | 120 |

### 2.4. Fiber Lasers Research Technique

The schematic diagram of the short-cavity laser experimental setup is depicted in Figure 3. A counter-propagating (or "backward") laser pumping was used. The laser cavity "1" was coupled via multiplexer filter (FWDM) "2" to the pump sources "3": 3S Photonics 1999CHP diode ($\lambda$ = 974.5 nm). In order to avoid heating of the active fiber due to the high absorption of the pumping radiation, it was fixed to a heat sink with the temperature

stabilized at 20 °C via Peltier (TEC) module [21]. The laser output radiation propagated through the corresponding port of the FWDM "2" to the fiber-optic isolator "4", which protected the optical circuit from reflected radiation. Additionally, FC/APC pigtail was spliced to the opposite end of the active fiber, at the side of the highly reflecting grating, to prevent the possible influence of reflected radiation on the stability of laser generation. The laser emission passed through the isolator was studied using one of the following devices: the Yokogawa AQ6370C optical spectrum analyzer ("5") with a maximum resolution of 0.02 nm and the CNI Laser PS100 optical power meter ("6").

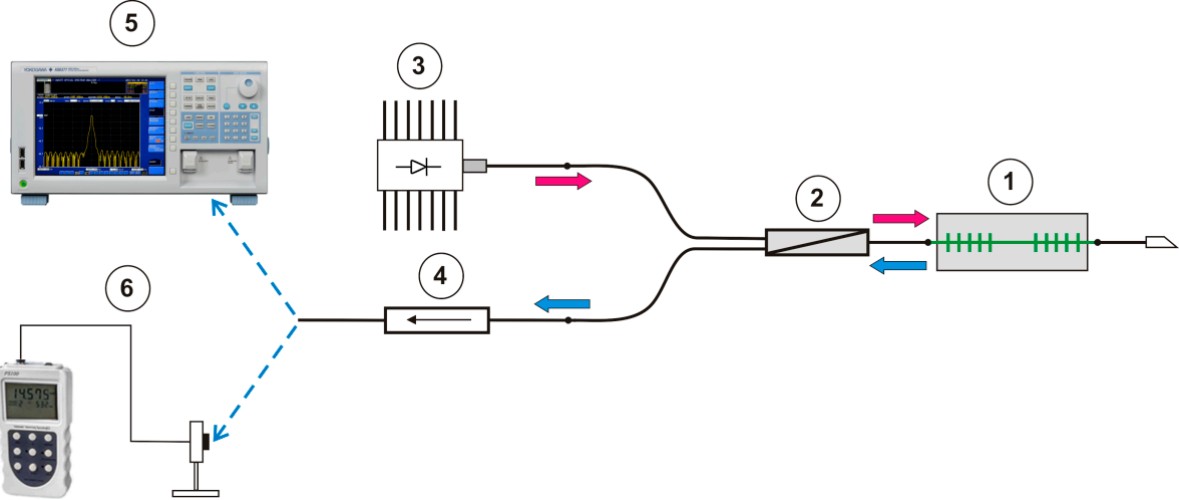

**Figure 3.** The schematic diagram of the laser investigation setup: 1—laser cavity; 2—fiber filter wavelength-division multiplexer (FWDM); 3—pump source; 4—fiber-optic isolator; 5—optical spectrum analyzer (OSA); 6—optical power meter. All fiber-optic elements in the laser scheme were interconnected by precise fusion splicing.

## 3. Results

This section presents the results of a study of the main characteristics of laser cavities "Laser #1" and "Laser #2". Let us first analyze the optical emission spectra of these resonators (Figure 4). It is noteworthy that the emission spectrum of "Laser #2", measured before the start of generation ("Laser #2, a"), shows peaks corresponding to neighboring longitudinal modes of the Fabry–Perot cavity. The distance between these peaks (~0.08 nm) corresponds to the intermode interval calculated using the technique from [19]. Once the generation threshold was reached, only one of the longitudinal modes was excited in the "Laser #1" and "Laser #2" cavities, as evidenced by the registration of a single narrow peak in the corresponding emission spectra: centered at a wavelength of 1065.08 nm (Figure 4, blue curve) and at a wavelength of 1066.38 nm (Figure 4, red curve). The width of each lasing peak, measured at a level of −3 dB from the intensity maximum, was ~0.04 nm, which is two times smaller than the longitudinal mode interval. Thus it can be concluded that both lasers were operated in the regime of generation of a single longitudinal cavity mode (single-frequency mode). The shift of the wavelength of the maximum radiation intensity in the spectra of both lasers by about −0.3 nm relative to the position of the peaks of the FBGs (Figure 4) is explained by the influence of the out-diffusion of molecular hydrogen from the fiber core.

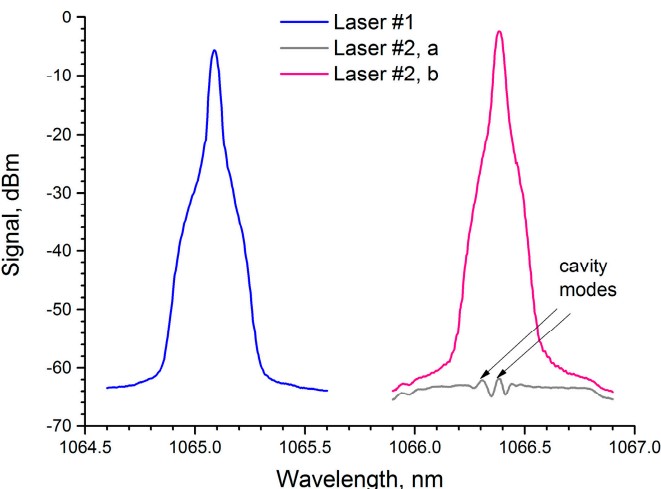

**Figure 4.** Emission spectra of the "Laser #1" and "Laser #2" cavity samples. The "Laser #2, a" and "Laser #2, b" spectra were measured before and after lasing threshold, respectively.

Based on the spectral curves in Figure 4, both of the studied lasers have the same emission linewidth: about 0.03 nm (full width half maximum, FWHM). It should be taken into account that this FWHM value is close to the maximal resolution of the used optical spectrum analyzer, and therefore, the actual emission linewidth of a Fabry–Perot short-cavity ytterbium fiber laser is much narrower—an order of several kHz [22,23]. However, the detailed characterization of the laser's emission cannot be processed via optical methods (it requires a special radio-frequency method, such as self-heterodyne) and, therefore, was beyond the scope of this work.

One of the important questions posed before starting this work was to determine the degree of influence of the composition of the glass matrix of the core of a ytterbium fiber on the laser efficiency of a short resonator fabricated on its basis. To this end, the dependences of the output laser radiation power on the pump radiation power were measured under identical conditions. Let us analyze and compare the corresponding dependencies presented in Figure 5. As can be seen, the threshold value of laser generation for the "Laser #1" cavity (10 mW) was an order of magnitude higher than for "Laser #2" (100 mW). The slope efficiency of Laser #1 (13%) was also significantly worse compared to Laser #2 (47%).

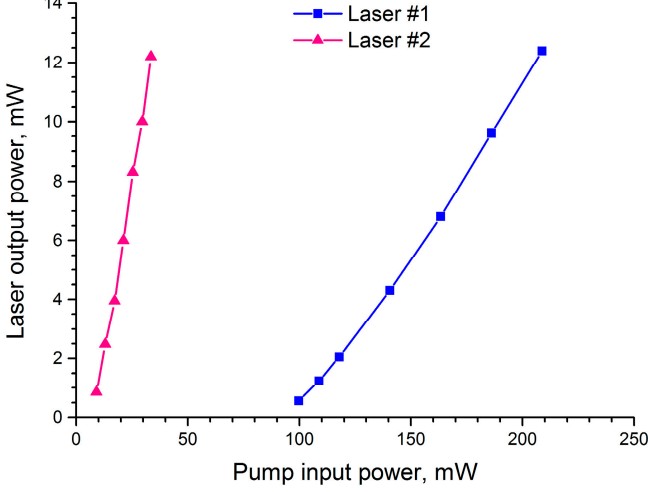

**Figure 5.** Laser optical power as a function of the pump power, measured for "Laser #1" (blue curve) and "Laser #2" (red curve) cavity samples.

## 4. Discussion

A comparison of the output characteristics of "Laser #1" and "Laser #2" revealed that the LD604 fiber is not as promising as the Y291 one from the point of view of using the fibers as an active element for single-frequency laser cavities. The main disadvantages of the "Laser #1" cavity, made on the basis of the LD604 fiber sample, are a relatively high lasing threshold as well as low slope efficiency. At the same time, according to its amplifying characteristics (pump-to-signal conversion efficiency), the LD604 fiber, based on the results of our previous work [8], was considered very promising. Degradation of this fiber could occur when photoinduced FBGs were inscribed in it. It is important to note that in "Laser #1" and "Laser #2" the FBGs had a total length of 16 mm, which was almost 90% of the total length of the cavities (18 mm). As a result, the pump radiation and the generated laser radiation inside these cavities propagated to a large extent over the core areas, which were subjected to intense UV irradiation with a photon energy of 6.4 eV. According to the results of [24,25], the impact of UV radiation with an energy of more than 5 eV on a glass network activated by ytterbium ions leads to a photoinduced transformation of a part of trivalent ions ($Yb^{3+}$) into the divalent state ($Yb^{2+}$), which do not participate in the process of laser generation. In addition, since before UV irradiation, the sections of the LD604 and Y291 fibers were saturated with molecular hydrogen, when FBGs were written in them, atomic-hydrogen-related photoinduced defects, including hydroxyl complexes, were induced in the core [16,26]. The population of induced hydroxyl centers and absorption bands associated with them increases with increasing UV dose [26]. Apart from additional optical losses near the pumping and emission wavelengths of ytterbium lasers caused by the absorption of hydroxyl centers, their presence in the nearest neighbors of active ions increases the probability of nonradiative relaxation of the excited state [27,28]. In other words, the action of UV radiation with a wavelength of 193 nm on hydrogen-loaded ytterbium fibers leads to a degradation of amplifying characteristics similar to its mechanism, which was previously found in irradiated erbium fibers [29,30].

As can be seen from Table 3, a significantly higher dose of UV radiation was spent to inscribe FBGs in the Laser #1 cavity compared to Laser #2. Thus, a significant decrease in the laser slope efficiency (by a factor of 3.6) and an increase in the lasing threshold (by an order of magnitude) in a laser sample fabricated on a segment of an LD604 fiber can be explained by three factors: transfer of ytterbium ions into the divalent state ($Yb^{2+}$); higher photoinduced losses that occur at the point where the FBGs are inscribed; a higher concentration of hydroxyl centers induced in the glass network and, consequently, a more probable nonradiative deactivation of excited $Yb^{3+}$ ions as a result of dipole–dipole interaction in them.

## 5. Conclusions

The optical and laser characteristics of highly doped ytterbium fibers fabricated by the MCVD method, with a core composition of $Yb_2O_3/Al_2O_3/P_2O_5/SiO_2$ and $Yb_2O_3/P_2O_5/SiO_2$ and with $Yb_2O_3$ content of 1.27 and 1.85 mol%, have been studied respectively.

Comparative studies of the photosensitivity of these fibers to UV irradiation from an excimer laser with a wavelength of 193 nm have been carried out. The inscribing of photoinduced FBGs with a high reflection coefficient (at least 99.9%) in these fibers is possible only in the presence of molecular hydrogen in the core glass network. It was found that the dynamics of FBG induced in fibers with a $Yb_2O_3/Al_2O_3/P_2O_5/SiO_2$ core is about five times slower than in fibers with a $Yb_2O_3/P_2O_5/SiO_2$ core.

Fiber laser cavities of the Fabry–Perot type were fabricated on segments of the studied ytterbium-doped fibers ~2 cm long, in which the functions of the highly reflecting and partially reflecting (outcoupling) mirrors were performed by wavelength-matched FBGs. Comparative studies of the dependences of the output power on the pump power at a wavelength of 974.5 nm were carried out in the fabricated cavities, which had the same effective length of ~5 mm. The highest slope efficiency (47%) was demonstrated by a laser cavity fabricated on a fiber segment with a core composition of $Yb_2O_3/P_2O_5/SiO_2$. The

lasing threshold in this sample was found at pump powers of 10 mW. For a cavity based on a fiber with a $Yb_2O_3/Al_2O_3/P_2O_5/SiO_2$ core composition, the slope efficiency turned out to be 3.6 times lower, only 13%, and the lasing threshold was an order of magnitude higher, approximately 100 mW. The relatively low lasing characteristics of a cavity based on a fiber with a $Yb_2O_3/Al_2O_3/P_2O_5/SiO_2$ core can be explained by the increased UV radiation dose required to induce the FBGs. Prolonged irradiation of the core saturated with molecular hydrogen led to the transformation of a part of the $Yb^{3+}$ ions into $Yb^{2+}$, excessive photoinduced losses at the sites of the FBGs and an increase in the concentration of hydroxyl centers induced in the glass network, upon interaction with which the excited $Yb^{3+}$ ions were nonradiatively deactivated.

Thus, as a result of the performed studies, two key factors were revealed which would make it possible to develop highly ytterbium-doped fibers with improved characteristics, optimized for use in the cavities of single-frequency fiber lasers with a lasing wavelength of ~1.06 μm—the composition of the core glass matrix and its photosensitivity to UV radiation. The output power of these lasers can be potentially increased to the hundred-milliwatts level by using more powerful pumping diodes or even to the watts level with the help of well-known MOPA (master oscillator and power amplifier) optical amplification circuits.

**Author Contributions:** Conceptualization, A.R. and D.L.; methodology, M.Y., A.A. and M.L.; software, A.R.; investigation, A.R., M.Y., A.A. and A.U.; data curation, A.R. and M.L.; writing—original draft preparation, A.R. and D.L.; writing—review and editing, A.U. and M.L.; supervision, D.L.; project administration, A.R. and D.L. All authors have read and agreed to the published version of the manuscript.

**Funding:** This research was funded by Russian Science Foundation, grant N 22-19-00511.

**Institutional Review Board Statement:** Not applicable.

**Informed Consent Statement:** Not applicable.

**Data Availability Statement:** Not applicable.

**Acknowledgments:** The authors are very grateful to O.V. Gryaznov and M.Yu. Artemyev from Optosystems Ltd., Moscow, Russia, for helpful consulting and for supporting the fiber Bragg gratings inscription setup.

**Conflicts of Interest:** The authors declare no conflict of interest.

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
