# Peer review of "Optimization of the Core Compound for Ytterbium Ultra-Short Cavity Fiber Lasers"

_fibers, doi:10.3390/fib11060052_

Round 1

Reviewer 1 Report

In the manuscript, the authors fabricated highly ytterbium, aluminum and phosphorus co‐doped silica fibers with low optical losses by the MCVD method. And the ultra‐short Fabry‐Perot laser cavities were fabricated by inscribing of two fiber Bragg gratings directly in the core of the fiber samples. This manuscript is in general well-written, but there are some questions need to be addressed.

1. The authors should highlight the importance of this work with a short discussion, considering the novelty here is moderate.

2. In the introduction, the author gives a comprehensive review of the relevant literature, but it is also necessary to clarify the advantages of the co‐doped silica fibers proposed in this manuscript.

3. am curious about how the authors obtain the parameters in table 2. Please clarify it.

4. More information should be provided to demonstrate the characteristics of the lasers, like Full Width Half Maximun, Side-Mode Suppression Ratio, ...

5. In line 17, the author described that With a help of hydrogen loading is was possibly to, I guess it is misuse of it as is, right?

Reviewer 2 Report

In this paper, ytterbium, aluminum, and phosphorus codoped silica fibers with low optical losses were fabricated by the MCVD method. By writing FBG on a phosphosilicate and aluminophosphosilicate glass Yb-doped fiber, the SFLs were achieved by these two different doped fibers. Compared with aluminophosphorosilicate fiber, the phosphosilicate fiber has a higher pump-to-signal conversion efficiency of about 47%. The investigation of the experimental analysis is thorough and complete. The manuscript could be published after addressing the following issues:

1. What is the technical difficulty of making the phosphosilicate and aluminophosphosilicate glass fiber, and how the cladding grows on the core?

2. Spelling mistake in line 67, the words “fiberssis” should be “fibers is”.

3. The ordinate of Fig. 1 should be changed to “Transmission”.

4. In Fig. 2, the authors should introduce the connect method between two different fibers in the laser system.

5. The FBG is sensitive to the environment. How did the laser output make stable when the temperature changed? Moreover, how to keep the wavelength of the two lasers working stable for a long time? Can the author give the measurement data in a long time?

6. Can the reflected wavelength of the two types of Yb-doped material (LD 604 and Y291) be on the same wavelength, and how does it?

7. By increasing the doping concentration of Laser #1, the low slope efficiency can improve or not. Why is this?

Reviewer 3 Report

The MCVD method was used to fabricate highly ytterbium, aluminum and phosphorus co-doped silica fibers with low optical losses. Hydrogen loading was used to induce highly reflective Bragg gratings, and the highest pump-to-signal conversion efficiency of 47% was achieved using phosphosilicate fiber. Here are my comments.

1. Para2 and Para4 are basically repeating the same information. I suggest the authors to make the text more concise and include other relevant information. 

2. Among the 25 references provided, the authors self-cited 10 of their own works. I suggest to minimize self-citation and include other reported works to strengthen the manuscript. 

3. I suggest the authors rephrase the following sentence in Page 2, "(Al2O3/P2O5/SiO2) has refractive index close to that of pure silica glass, which is coursed by formation of AlPO4 join from Al2O3 and P2O5 structure units [12] and the main impact to the core refractive index is due to refractivity of Yb2O3."

4. In Section 2.1, it is suggested to make a figure out of Para2 to help readers grasp the process of core glass fabrication. 

5. In Table 2, The Loss at 975 nm in dB/m is 2600 and 1700. Are these numbers correct?

6. In Section 3 of Results, authors mentioned the influence of out-diffusion molecular hydrogen from the fiber core. How did this happen and what can be done to overcome this issue?

7. In Section 5 of the conclusions, effective length was mentioned when this was not discussed in other sections of the manuscript. The authors should elaborate further on how this was determined. 

8. The authors misspelled phosphosilicate and aluminosilicate in the abstract. 

Round 2

Reviewer 1 Report

I think the manuscript has been sufficiently improved to warrant publication in Fibers. 

Author Response

Thank you for your valuable comments.

In this article we have shown clearly that the best choice for single-frequency laser fabrication is the heavily ytterbium-doped phosphosilicate fiber. Compared to phosphate fibers, the developed fiber can be easily integrated (spliced with a low-loss level) with conventional fiber-optic elements. This fiber had also a relatively good UV-photosensitivity, IR-photodarkening-stability and signal gain properties, which allowed us to demonstrate the single-frequency laser sample with slope efficiency (pump-to-signal conversion) of 47 % at the 1066.4 nm lasing wavelength.  In a future research, this laser may be applied as a source ("seed") of narrow-band reference radiation for the second (or even fourth) garmonic generation purposes, similar to well-known Nd:YAG solid-state laser. Unlike a solid-state laser, a fiber laser does not require powerful cooling, consumes little energy, and takes up little space.